# Motorization and its consequences: A mixed-methods study on the epidemiology, impact, and stakeholder perspectives of road traffic injuries among undergraduate two-wheeler riding students in Nepal

Nikita Bhattarai[1,2]*, Diksha Pokhrel[1], Pranil Man Singh Pradhan[3], Sunil Kumar Joshi[1]

1 Department of community medicine, Kathmandu Medical College, Kathmandu, Nepal, 2 Current affiliation: NIHR Global Health Research Center for Multiple Long-Term Conditions, Kathmandu Medical College, Kathmandu, Nepal, 3 Department of Community Medicine and Public Health, Maharajgunj Medical Campus, Institute of Medicine, Tribhuvan University, Kathmandu, Nepal

* nikitanihr@kmc.edu.np

## Abstract

### Introduction

Motorization has undeniably enhanced mobility and convenience, but it comes at a significant cost, as the increasing number of vehicles has led to a surge in road traffic injuries (RTIs) now a major global cause of disability and death. Over the past decade, the significant growth in two-wheelers has coincided with a rise in RTIs, creating an increasing concern for young individuals. This mixed-method study aims to explore epidemiology, assess their impact on undergraduate students, and gather key recommendations from stakeholders to improve road safety and reduce injury rates.

### Methodology

The quantitative data collection was conducted among undergraduate students in the four colleges in Kathmandu district. The total number of participants in this study was 1217 students of 17–31 years of age. Furthermore, for the qualitative data collection, 5 Key informant interviews were conducted among various stakeholders.

### Results

The prevalence of RTI in 2021–2022 was 23%. The most common injuries were bruises in the upper and lower limbs. It was seen that only 78% of the respondents had a driving license. The injuries led to academic loss and caused financial strain due to healthcare and vehicle repair expenses. 5 key recommendations to decrease the ever-increasing burden of RTI have emerged from the qualitative data analysis.

**Data availability statement:** The minimal anonymized dataset necessary to replicate the findings of this study has been deposited in the Zenodo repository. It is freely accessible at https://doi.org/10.5281/zenodo.18204329.

**Funding:** The study was financially supported by the Nepal Health Research Council through the "Post Graduate Health Research Grant, 2020" (Grant Number: 1585), awarded to the corresponding author, Nikita Bhattarai (NB) for her master's thesis. The funder had no role in study design, data collection, analysis, decision to publish, or preparation of the manuscript.

**Competing interests:** The authors have declared that no competing interests exist.

## Conclusion

The study highlighted a high prevalence and impact of RTIs among undergraduate students along with 5 key recommendations that came forward from this study could be instrumental for road safety in Nepal.

## Introduction

Road traffic injury (RTI) is defined as physical damage sustained by a person as a result of a road traffic incident/Crash [1]. Road crashes and deaths are becoming a matter of serious concern and a major public health problem with the increasing number of deaths, hospitalizations, and disabilities, especially among the productive young population [2]. RTIs cause widespread harm, ranking ninth globally in disability-adjusted life years (DALY) lost [3].

According to WHO, Motorcycles, scooters, and e-bikes are at higher risk of RTI, accounting for **30%** of all road traffic deaths in **2022** [4]. Injuries are disproportionately high among males, disadvantaged groups, and young adults aged **15–29 years**, particularly in road crashes, where disadvantage groups include low-income individuals, informal workers, rural residents, where they face elevated risk due to limited access to safety infrastructure, protective gear, legal protections and emergency services [4]. Over the past decade, Nepal has seen a dramatic increase in vehicle registrations [5]. In **2011**, registrations surged by **325%**, with **85%** of newly registered vehicles being two-wheelers. By **2021**, motorcycle registrations reached an all-time high of **556,819** units, reflecting the growing preference for two-wheelers as a primary mode of transport. This trend highlights the increasing accessibility and demand for personal vehicles, particularly motorcycles, across the country. [6,7]

Despite road safety data being gathered from multiple sources, including police reports, traffic records, insurance databases, and hospital reports, under-reporting remains a significant issue, particularly in Nepal and India, limiting a comprehensive understanding of the true burden of RTIs [8–10]. The primary sources, traffic police records and hospital data, often fail to capture the full extent of RTIs due to limitation in epidemiological, economic, and risk factors [11]. As a result, RTIs remain inadequately recognized as a critical public health issue, with gaps in data visibility hindering meaningful policy interventions [10,11]. This mixed-method study aims to assess the prevalence, risk factors and underlying mechanisms of RTIs among undergraduate two-wheeler drivers in Kathmandu. Additionally, it seeks to evaluate health-seeking behaviors, estimate the broader impacts of RTIs, and identify key recommendations derived from qualitative analysis.

## Methodology

The quantitative study was conducted in four Kathmandu University-affiliated colleges in Kathmandu from December 2020 to February 2021. The ethical approval was obtained from Nepal Health Research Council (289) and the Institutional Review Committee of Kathmandu Medical College, reference number (1009210917) The

permissions were taken from the principal, program coordinators and from the administration department to obtain the list, the phone numbers, and email addresses of the students and the college informed the students regarding the study. The data were collected from students enrolled in first to fourth year, spanning 28 classes from 4 academic institutions: Kathmandu Medical College, Nepal Medical College, National college and St. Xavier's College.. It covered diverse streams including Medicine, Dental, BSc Nursing, Development Studies, Finance, and bachelor's in social work.

The questionnaire was emailed to all 1,879 students across the 28 classes with two reminders sent weekly for two weeks. After two weeks the survey was closed. The participants gave online informed consent and participation in the study was voluntary.

All the students who use a two-wheeler irrespective of driving license currently studying in four colleges were included in the study and those participants who ride two-wheelers who did not give consent and cycle riders were excluded.

For the qualitative data 5 key informant interviews (KII) were conducted including the senior college program coordinator, emergency physician, traffic police officer, former Director-general of the Department of transport management, and judge They were informed about the study, and it was recorded only after they gave written consent. The interview was conducted in the Nepali language and lasted about 15–20 minutes. The qualitative component of this study was explicitly designed to elicit recommendations for reducing RTI from key stakeholders. The interview guide was structured around this purpose, and participants were asked to provide their perspectives on actionable strategies.

This study employed explanatory sequential mixed-methods design. In the first phase, quantitative data was collected and analyzed to determine the prevalence of RTI and associated risk factors. The second, qualitative phase was informed by these findings and involved five key informant interviews to generate contextual recommendations to reduce RTI. The integration of both strands allowed quantitative results to establish the burden of RTIs, while qualitative insights provided stakeholder perspectives and actionable strategies.

### Data analysis

**Quantitative data:** Descriptive statistics, including frequency, percentage, median, IQR, and graphs, were used to present the results. Pearson chi-square and Fischer exact tests were applied for bivariate analysis, with COR at a 95% CI to assess associations. Multivariate binary logistic regression adjusted variables with $p < 0.02$ to calculate the AOR.

**Qualitative data:** Qualitative data from KIIs was transcribed verbatim into Nepali, reviewed for discrepancies, and analyzed using thematic analysis. Meaningful units were identified, condensed, labeled as codes, and sorted and themes were formulated with translated verbatim quotes presented in italics.

The data analysis was done in SPSS and NVivo respectively.

### Results

The total number of students studying in these colleges was 18,798, out of which 1,217 responded, giving a response rate of 64.8%. Among the 1,217 respondents, 579 (52.4%) used two-wheelers. The participants ranged from 17 to 31 years and 308 (65.6%) were male respondents. The most used two-wheeler was motorcycle (330, 57.0%) followed by scooter (249, 42.7%) (Table 1).

Out of 579 two-wheeler users, 218 (37.6%) reported road traffic incidents and among them, 49 (31%) had multiple incidents, and 19 (3.2%) experienced more than three. However, only the most recent incident per respondent was considered, regardless of severity. Not all road traffic incidents necessarily led to RTI; when enquired further, only 135 (23.3%) out of 218 incidents resulted in RTI. Among the 135 respondents who sustained injuries, 79(58.5%) were male and 56(41.5%) were female. These injury details corresponded only to two-wheeler drivers and did not include pillion passengers. Pillion riders were present in 139 (24%) of incidents, and 99 (71%) sustained minor injuries, but no fatalities were recorded.

October was the most common month for incidents (40, 18.3%), while the majority occurred over weekends (74, 33.8%), nearly half (107, 49.3%) between 12 PM and 6 PM, and (98, 45%) on the ring road, particularly in Chabahil,

**Table 1. Age, institution and driving experience and license details of the undergraduate students who use two-wheelers (n = 579).**

| Variable | Frequency | | Variable | Frequency | |
|---|---|---|---|---|---|
| | **N** | **%** | | **N** | **%** |
| **Age grouping** | | | **Institutions** | | |
| < 19 years | 47 | 8.1 | Kathmandu Medical College | 241 | 41.6 |
| 20-22 years | 308 | 53.2 | Nepal Medical College | 176 | 30.4 |
| 23-25 years | 213 | 36.8 | St. Xaviers College | 41 | 7.1 |
| >25 years | 11 | 1.9 | National College | 121 | 20.9 |
| **Gender** | | | **Type of two-wheeler** | | |
| Male | 308 | 65.6 | Motorcycle | 330 | 57.0 |
| Female | 199 | 34.4 | Scooter | 249 | 43.0 |
| **Duration of their driving experience** | | | **Duration of acquiring driving license** | | |
| | **N** | **%** | | **N** | **%** |
| | | | Driving without a driver's license | 127 | 21.9 |
| < 1 year of driving experience | 128 | 22.1 | Driving with < 1 year of acquiring driving license | 185 | 32.0 |
| 1-3 years of driving experience | 265 | 45.8 | Driving with 1–3 years of acquiring driving license | 223 | 38.5 |
| 4-6 years of driving experience | 130 | 22.5 | Driving with 4–6 years of acquiring driving license | 40 | 6.9 |
| 7-10 years of driving experience | 56 | 9.7 | Driving with 7–10 years of acquiring driving license | 4 | 0.7 |

Koteshwor, and Gwarko. Collisions accounted for 119 (54.4%) of incidents, with 100 (45.6%) involving another two-wheeler. Speeding (76, 34.8%) and sudden braking by another driver (47, 21.7%) were leading causes. Non-collision incidents (99, 45.6%) were mainly due to vehicle slip (48, 48.6%) and imbalance (45, 45.8%). Mechanical defects in the two-wheeler were reported by 18 (8.3%) of respondents.

Out of the 135 RTIs, 769 injuries were recorded as some respondents sustained multiple injuries. Lower limbs were most affected, including the knee (91, 11.8%) and lower leg (90, 11.7%), while elbow injuries (77, 10%) were common in upper limbs. Bruises (328, 42.6%) were most frequent, followed by sprains (192, 25%), fractures (75, 9.7%), and joint dislocations (39, 5%).

Out of all incidents, more than half of respondents (139, 64%) required treatment. Nearly half (102, 47%) opted for the emergency room. Of the 11 respondents who needed further management, 9 (84.6%) required admission, and 6 (53.5%) were admitted for 5 days, with the maximum duration being 1 week.

More than half of respondents (91, 57%) reported disability after the injury. The majority (118, 74.5%) could not perform regular activities for less than 1 week.

Financial impacts were also evident. Half of the respondents (79, 50%) paid up to NPR 1,000 for treatment, mostly for bruises. Nearly (14, 9%) paid up to NPR 50,000 for more serious injuries. The healthcare cost ranged from NPR 200–70,000, with a median of NPR 1,350 and IQR of NPR 4,425. Insurance covered costs for 47 (30%), whereas 108 (70%) paid out of pocket. In addition, 58 (40%) incurred repair costs for their two-wheelers, ranging from NPR 1,000–125,000.

As undergraduate students, respondents also experienced educational impacts. (96, 66.6%) took leave for less than 1 week, whereas (27, 19%) took leave for more than 2 weeks. The median and IQR for leave days were 7 and 10 days respectively, with a range of 1–28 days. (2, 9.6%) missed examinations due to injuries.

## Association between respondent's attributing factors with RTI among two wheelers using bivariate and multivariate analysis

The study identified key factors associated with RTIs. (Table 2) The Crude odd's ratio demonstrated that males were more likely to sustain RTIs than females, and scooter users demonstrated a significant association with RTIs. Riders with 4–6 years of driving experience were at higher risk. All the associations were found to be statistically significant. It was followed by multivariate analysis.

**Table 2. Adjusted odds ratio with various variables of the respondents with RTI (n = 579).**

| Variables | Unadjusted Odds | | Adjusted Odds | |
|---|---|---|---|---|
| | COR [95% CI] | P value | AOR [95% CI] | P value |
| Gender | | | | |
| Male | 1.4[1-2.2] | **0.04*** | 0.98 [0.4-1.2] | **0.04*** |
| Female | RC | – | RC | – |
| Type of two-wheeler | | | | |
| Motorcycle | 1.3[0.86-1.9] | 0.02 | 1.9[1.2-3.1] | **0.008*** |
| E. scooter | 3.9[0.23-62] | 0.34 | 6.5[0.38-109] | 0.19 |
| Scooter | RC | – | RC | |
| Years of driving duration | | | | |
| < 1 year | 1.1[0.55-2.3] | 0.74 | 0.95[0.39-2.5] | 0.95 |
| 1-3 years | 1 [0.5-2] | 0.83 | 0.75[0.35-1.9] | 0.45 |
| 4-6 years | 0.4[0.19- 0.94] | **0.03*** | 0.4[0.15-0.92] | **0.01*** |
| 6-10 years | RC | – | RC | – |

* Statistically significant, RC: Reference Category, CI: Confidence Interval, COR: Crude Odds Ratio AOR: Adjusted Odds Ratio, adjusted for age and sex.

The multivariate analysis highlighted that male had higher odds of RTIs compared to females, and motorcycles were significantly associated with increased risk. Riders with 4–6 years of experience showed reduced odds.

The qualitative section was complemented by 5 key informants' interview were conducted with experienced stakeholders including senior college program coordinator, emergency physician, traffic police officer, former Director-general of the Department of transport management, and judge. These interviews were specifically designed to gather expert recommendations on reducing RTIs. A major strength of this study lies in the depth of insights provided by the informants, whose perspectives have been refined into 5 overarching themes as summarized in Table 3.

Together, these themes highlight a multi-layered approach to road safety: early education to instill discipline, infrastructure and transport reforms to reduce risk exposure, stronger enforcement by traffic police, legal reforms to deter risky behaviors, and licensing reforms to ensure competency. The verbatim quotes provide the rationale behind each recommendation, grounding them in stakeholder perspectives and reinforcing their relevance to policy and practice.

## Discussion

This study examined RTIs among undergraduate two-wheeler users in Kathmandu. Most RTIs occurred between 12 PM and 6 PM, aligning with patterns observed in Nepal and India [12–14]. The time frame coincides with peak traffic hours, and key informants including traffic police and emergency physicians confirmed that the two-wheeler incidents are frequently reported during these hours, reinforcing the need for time targeted enforcement and awareness.

Injuries between 6–10 PM may be linked to driver fatigue, as seen in studies from China and Croatia [15,16]. Weekend incidents were more frequent, possibly due to increased travel activity, speeding and reduced traffic congestion [14–16]. This pattern is consistent emergency room data from Nepal [17,18] as with global trends as mentioned by WHO [2].

RTI concentration in October likely coincides with festive travel during Dashain and Tihar [19] which highlights the needs for different traffic management strategies around those months. Informants emphasized that festive periods bring increased mobility and risk suggesting that traffic police should intensify monitoring and during these months.

**Table 3. The themes and recommendations from the qualitative study.**

| Themes | Description of the themes | Verbatim |
|---|---|---|
| **Themes 1**: Education and awareness for change of driving behavior | Stakeholders emphasized the importance of early education in shaping safe driving practices. They recommended introducing road safety awareness programs at the higher secondary level and incorporating traffic discipline into the pre-primary curriculum to instill long-term behavioral change. In addition, integrating road safety modules into college orientation programs and conducting regular awareness campaigns targeting young riders were suggested as strategies to gradually shift driving behavior. | *"If the Pre-Primary or the primary level curriculum includes road safety discipline, children will be conditioned to follow road safety. The change might take 15-16 years to take place, but it will happen"- **Senior Program coordinator**.* |
| **Theme 2**: Decrease dependency on Two-wheelers | Participants stressed that reducing reliance on two-wheelers requires both infrastructure improvements and better public transport. Road maintenance was identified as a priority, alongside strengthening and modernizing public transportation systems to make them efficient, reliable, and well-equipped. Fiscal measures such as two-wheeler and pollution taxes were also proposed to discourage excessive use of motorcycles and scooters, while integrated transport systems were seen as essential to provide safer alternatives. | "When *Public transportation becomes efficient and if it has facilities, it can attract many people. The priority is to make public transportation effective followed by the increment of taxes such as Two-wheeler tax and pollution tax "–* **Ex-Director General of DOTM** |
| **Theme 3**: Role of Traffic Police | The role of traffic police was highlighted as central to enforcement and behavior change. Recommendations included increasing police allowances to improve morale, strictly enforcing speed limits through random checks, and discouraging political interference in policing decisions. Equipping traffic police with real-time license verification technology was also emphasized to strengthen monitoring and reduce fraudulent practices. Overall, stakeholders believed that consistent enforcement and adequate resources could significantly reduce violations and crashes. | *"The rule of drinking and driving when it was launched in by Nepal police. They tried to control it and it got controlled in a few years. It's a success story."* **Judge** |
| **Theme 4**: Legal system | Legal reforms were considered essential to strengthen compliance and deter risky behaviors. Recommendations included enforcing mandatory helmet use for both drivers and pillion riders, imposing proportionate fines and punishments based on the severity of negligence, and raising the minimum age for a driving license to 20 years. Stronger penalty systems, including suspension of licenses for repeated violations, were seen as more effective than compensation alone in changing behavior among young riders. | *"The age group 18–24 will not listen to anyone, so we need to lobby for a stronger penalty system. People won't change their behavior unless you can link their actions to suspension. "**Senior Program coordinator.*** |
| **Theme 5**: License system | Stakeholders advocated reforms in the licensing process to ensure competency and accountability among riders. Suggestions included mandatory road tests, provisional licenses with supervised driving periods, and revocation of licenses for violations during trial phases. Continuous improvements in written and practical examinations were also emphasized to strengthen the licensing system and ensure that only eligible riders qualify. These measures were seen as critical to reducing crashes and instilling discipline among new drivers. | *If driving license should be given under a trial period for a year, always be accompanied by someone while driving for a year before being upgraded to a full license. If the rider drives during this period to break a traffic rule or gets involved in an incident, his license should be revoked."* **Emergency Medicine Physician** |

Collision data revealed that 46% of incidents involved another two-wheeler and was found similar in studies in Kashmir India but slight less was found in another study done in Nepal [20,21], reinforcing the vulnerability of motorized two-wheelers in urban traffic ecosystems. Informants consistently attributed this to reckless driving and poor adherence to traffic rules, particularly among young riders. Their recommendations included road safety education into school curriculum and launching targeted campaigns over time to reinforce the emphasis on human behavioral factors.

Mechanical defects, though reported by only 8.3% of riders, mirror findings from Kathmandu-based studies [21–22] and underscore the importance of regular vehicle inspections. Informants highlighted gaps in vehicle maintenance and called for stricter inspection protocols. These align with WHO's on two-wheeler safety which emphasizes on vehicle conditions and enforcement of vehicle standards and regular inspection [4].

Injury patterns showed the upper limb was most affected (49.9%), followed by lower limb injuries (39.4%), also found consistent with studies in Nigeria, Nepal, and Laos [23–25]. The predominance of bruises (42.6%) and sprains (25%) [26–29] suggests that while many injuries are minor, their cumulative impact on academic performance and financial

stability is substantial. Nearly 13.3% of respondents took leave from college post-injury and treatment costs ranged from NPR 1,000 to NPR 50,000, with 60% paying out-of-pocket. Notably, vehicle repair costs often exceeded healthcare expenses, this economic strain is echoed in studies from Vietnam, Bangladesh and Nepal [30–33]. WHO has also mentioned that economic impact is disproportionally high pushing families into poverty due to high out of pocket expenses [4].

Strengthening the role of traffic police was also seen as a critical point. The informants recommend random license checks, stricter enforcement of speed limits and equipping officers with real- license verification tools. They also stressed the importance of political interference to ensure consistent application of rules for everyone. These suggestions align with the best global practices outlined in WHO road safety manual [1,3] and road safety charter and European transport safety council (ETSC) [34–36].

Legal reforms were also found essential as suggestions included increasing age limit for license eligibility, implementation of suspension for violations and enforcing use of helmet use for both riders and passengers. Informants argued that the penalties should be proportionate to severity of negligence. These suggestions align with the WHO recommendations for graduated licensing system and stricter penalties for risky behaviors [4]. The licensing system was also identified as areas where reforms were necessary. They advocated for a learner's permit system that included a supervised driving period before attaining full license, these again align with the global best practices [4].

The quantitative findings of this study highlighted the scale and nature RTI among two-wheeler users in Kathmandu. Patterns in injury mechanisms, types of injuries sustained, and the duration of license holding and driving experience revealed important risk factors. These data also underscored the significant health and financial burden, with high out-of-pocket expenditures, frequent disability, and disruption of education. Such evidence provides a strong basis for targeted intervention and improved post-crash care, ensuring that the burden of RTI doesn't disproportionately fall on students and their household.

The qualitative insights complemented these findings by offering stakeholder-driven recommendations. Informants emphasized shared responsibility across road users, policymakers, enforcement agencies, and the health system, aligning with global Safe System principles. [5] They called for reducing dependency on two-wheelers through stronger public transportation, fiscal measures such as pollution and vehicle taxes, stricter enforcement of traffic laws, and reforms in the licensing system. Together, these perspectives highlight that addressing RTIs among young riders requires both data-driven interventions and systemic reforms, ensuring safer roads, equitable health financing, and coordinated multi-sectoral action

## Conclusion

The integration of qualitative insights with quantitative findings offers a comprehensive understanding of RTI landscape. Together, they contextualize the data and illuminate challenges that are common across many low- and middle-income countries, such as high rates of motorcycle registration, underreporting of RTIs, and the resulting academic, economic, and health consequences for young populations. Quantitative analysis revealed a significant prevalence of RTIs among undergraduate two-wheeler riders, even during the pandemic, with notable disruptions to education and financial stability. Informants proposed a range of interventions such as graduated licensing, stricter penalties, enhanced enforcement, and regular vehicle inspections that do align closely with WHO guidance and global best practices. However, as seen in Nepal and elsewhere, implementation remains inconsistent, and the persistent gap between evidence and action underscores the urgency of translating proven strategies into sustained, context-sensitive interventions that reflect local realities and priorities.

## Recommendations

Strengthening hospital injury registries and linking them with police crash reports is essential to guide policy and capture injury mechanisms, severity, and rider experience. Equally important are affordable trauma care, physiotherapy

and rehabilitation centers and services integrated into student health programs to reduce long-term disability, alongside expanded health insurance to ease the financial burden on young riders and their families. While these measures are well recognized globally, their integrated implementation remains limited in many low- and middle-income countries, where fragmented efforts fail to address the combined health, financial, and academic impacts of RTIs. A holistic approach is therefore necessary by tailoring strategies based on local realities, with a focus on preventing disability and reducing financial burden among students, is critical to ensure that road safety measures to deliver equitable, sustained impact.

## Limitations of the study

This study faced limitations, including potential non-responses due to issues with email delivery via Google Forms, as some of the email went into spam folders and some of the email addressed provided by the administration was not being used by the participants, recall bias affecting respondents' injury details, and restricted generalizability as it was limited to one district in Kathmandu Valley.

## Supporting information

**S1 Data. Data RTA.**
(XLSX)

## Acknowledgments

We would like to thank all the undergraduate who contributed to our studies as participants, students along with Dr. Nishchal Dhakal, Mr. Ujjwal Upadhyay, Father Augustine Thomas for providing me the necessary support to conduct the study in their respective colleges. We would also like to thank the 5 key informants who provided their valuable time and experience without whom the research would not have been completed.

## Author contributions

**Conceptualization:** Nikita Bhattarai, Sunil Kumar Joshi.

**Formal analysis:** Nikita Bhattarai.

**Investigation:** Nikita Bhattarai.

**Methodology:** Nikita Bhattarai.

**Project administration:** Nikita Bhattarai.

**Resources:** Nikita Bhattarai.

**Supervision:** Sunil Kumar Joshi.

**Visualization:** Nikita Bhattarai.

**Writing – original draft:** Nikita Bhattarai.

**Writing – review & editing:** Diksha Pokhrel, Pranil Man Singh Pradhan, Sunil Kumar Joshi.

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
