## [Decision Letter · Decision Letter 0]

24 Sep 2025

PONE-D-25-42947Motorization and Its Consequences: A Mixed-Methods Study on the Epidemiology, Impact, and Stakeholder Perspectives of Road Traffic Injuries Among Undergraduate Students in NepalPLOS ONE

Dear Dr. Bhattarai,

Thank you for submitting your manuscript to PLOS ONE. After careful consideration, we feel that it has merit but does not fully meet PLOS ONE’s publication criteria as it currently stands. Therefore, we invite you to submit a revised version of the manuscript that addresses the points raised during the review process.

We look forward to receiving your revised manuscript.

Kind regards,

Satabdi Mitra, M.D(Community Medicine )

Academic Editor

PLOS ONE

Journal Requirements:

We would like to thank all the undergraduate who contributed to our studies as participants, students along with Dr. Nishchal Dhakal, Mr. Ujjwal Upadhyay, Father Augustine Thomas for providing me the necessary support to conduct the study in their respective. We would also like to thank the 5 key informants who provided their valuable time and experience without whom the research would not have been completed. We would like to acknowledge Nepal Health Research Council for providing the financial support by awarding the "Post Graduate Health Research Grant, 2020".

The funding was awarded by Nepal Health Research Council for PG health research grant 2020 to the corresponding author, Nikita Bhattarai (NB) for her master thesis. The funder has no role in study design, data collection, analysis, decision to publish nor preparation of the manuscript

5. Please amend your list of authors on the manuscript to ensure that each author is linked to an affiliation. Authors’ affiliations should reflect the institution where the work was done (if authors moved subsequently, you can also list the new affiliation stating “current affiliation:….” as necessary).

6. We note you have included a table to which you do not refer in the text of your manuscript. Please ensure that you refer to Tables 1 and 23 in your text; if accepted, production will need this reference to link the reader to the Tables.

8. Please remove all personal information, ensure that the data shared are in accordance with participant consent, and re-upload a fully anonymized data set.

Reviewers' comments:

Reviewer's Responses to Questions

**Comments to the Author**

1. Is the manuscript technically sound, and do the data support the conclusions?

Reviewer #1: Partly

Reviewer #2: Yes

2. Has the statistical analysis been performed appropriately and rigorously? 

Reviewer #1: Yes

Reviewer #2: Yes

3. Have the authors made all data underlying the findings in their manuscript fully available?

Reviewer #1: Yes

Reviewer #2: Yes

4. Is the manuscript presented in an intelligible fashion and written in standard English?

Reviewer #1: Yes

Reviewer #2: Yes

5. Review Comments to the Author

Reviewer #1: Title – acceptable but title says about all students but result is only for two-wheeler drivers? change the title or add the data of pillion riders also if you have the data.

Methods – is there secondary records of the police system and triangulate with your prevalence of RTI? As the prevalence you claim is lower 23%

Results – a response rate of 64.8% is very low and your results may be biased – justify

, 6.3% were under the influence of alcohol -is this self-reported? or was it confirmed by some agency?

Discussion – line 129 -lower limbs were most affected, but in line 190 -Injury patterns showed the upper limb was most affected (49.9%), is this from different study or the narration is inconsistent.

Recommendations- are general – what u can say (my suggestions only) is drive for better licencing program, proper multi-level data capture for RTA and RTI.

Most important

What are the infrastructure data you have collected for RTA? road banking, road marking etc.

Environmental factors – rain/fog/road repair / road illumination at a particular place,

You can use some CI for proportions presented.

Some good data is with this team needs proper presentation.

Reviewer #2: REVIEW

Motorization and Its Consequences: A Mixed-Methods Study on the Epidemiology, Impact, and Stakeholder Perspectives of Road Traffic Injuries Among Undergraduate Students in Nepal

Over all: This study is relevant as it addresses the rising burden of road traffic injuries among youth in Nepal, a vulnerable yet often overlooked population in public health policy. With the suggested changes, it can be considered for publication.

Abstract

Line 29: Methodology: to check this statement ’Methodology: The quantitative section was conducted among undergraduate students in the four colleges in Kathmandu district.’ Quantitative section?

Introduction:

Line 52 and 54: ‘Injuries are disproportionately high

among males, disadvantaged groups,’ add why they are disadvantaged groups.

Line 77, 78: It is mentioned 28 classes, Specify what are classes. ‘ Participation was voluntary and gave informed consent ‘ check the grammar of this statement

Methodology: Add timeline/ duration of the study

Results: Line 168, Check the grammar. Table 23 appears before table 4. Where are table 2 and 3?

Discussion:

Line 198 and 199: ‘no statistically significant association

was found between licensing status and RTI occurrence’. This is not reflected in results.

‘The themes and recommendations from the qualitative study’ should be better discussed in this section.

The discussion could be strengthened further through more robust comparisons with national and international studies, which would help contextualize the findings, highlight unique challenges in the Nepali context, and support evidence-based recommendations.

6. PLOS authors have the option to publish the peer review history of their article (what does this mean?). If published, this will include your full peer review and any attached files.

Reviewer #1: **Yes:**Dr Shrinivasa B Marinaik

Reviewer #2: No

---

## [Author Response · Author response to Decision Letter 1]

26 Nov 2025

Please find all the point by point response to the reviewer’s comments below:

Reviewer #1 Comments

Reviewer #1: Title – acceptable but title says about all students but result is only for two-wheeler drivers? change the title or add the data of pillion riders also if you have the data.

• The title has been changed to Motorization and Its Consequences: A Mixed-Methods Study on the Epidemiology, Impact, and Stakeholder Perspectives of Road Traffic Injuries Among Undergraduate Two-Wheeler riding Students in Nepal. To make it more clear as only the undergraduate students who were using a two wheeler were considered to be a part of the study.

Methods – is there secondary records of the police system and triangulate with your prevalence of RTI? As the prevalence you claim is lower 23%

• This study was done during the pandemic hence the prevalence of the study seems to be lower than the secondary records of the police system

Results – a response rate of 64.8% is very low and your results may be biased – justify

, 6.3% were under the influence of alcohol -is this self-reported? or was it confirmed by some agency?

• The response rate was relatively low due to several logistical challenges. Some of the email addresses provided by the colleges were inaccurate, while others belonged to students who no longer actively used those accounts. Additionally, a portion of the survey invitations may have been filtered into recipients’ spam folders, further limiting visibility and participation.

• The 6.3% of the participants who were under the influence of alcohol was self-reported.

Discussion – line 129 -lower limbs were most affected, but in line 190 -Injury patterns showed the upper limb was most affected (49.9%), is this from different study or the narration is inconsistent.

• I reviewed the section in question. The statement in line 129, indicating that the lower limb was most commonly affected, reflects the findings of my own study. In line 190, the reference to two studies from Nepal, one from Nigeria, and one from Laos also supports this pattern, showing consistent results across different settings. The discrepancy was due to a typing error, which I’ve now corrected.

Recommendations- are general – what u can say (my suggestions only) is drive for better licensing program, proper multi-level data capture for RTA and RTI.

Most important

What are the infrastructure data you have collected for RTA? road banking, road marking etc.

Environmental factors – rain/fog/road repair / road illumination at a particular place,

You can use some CI for proportions presented.

• Thank you for your thoughtful feedback. The recommendation for better licensing program has been included in the manuscript. The environmental factors and driving behaviors were indeed explored as part of the original thesis, but they are being developed into a separate manuscript due to space constraints in the current paper. This study was already quite extensive, so we’ve planned a follow-up publication that will specifically address infrastructure, environmental influences, and behavioral aspects in more detail. I also appreciate your suggestion regarding recommendations and will incorporate more targeted and actionable points as advised.

Some good data is with this team needs proper presentation.

Reviewer #2: REVIEW

Motorization and Its Consequences: A Mixed-Methods Study on the Epidemiology, Impact, and Stakeholder Perspectives of Road Traffic Injuries Among Undergraduate Students in Nepal

Over all: This study is relevant as it addresses the rising burden of road traffic injuries among youth in Nepal, a vulnerable yet often overlooked population in public health policy. With the suggested changes, it can be considered for publication.

Abstract

Line 29: Methodology: to check this statement ’Methodology: The quantitative section was conducted among undergraduate students in the four colleges in Kathmandu district.’ Quantitative section?

• The methodology section has been corrected from section to data collection

Introduction:

Line 52 and 54: ‘Injuries are disproportionately high

among males, disadvantaged groups,’ add why they are disadvantaged groups.

• This line is cited from WHO Powered Two- and Three-Wheeler Safety Manual (2022), and it refers to populations who face systemic barriers to safety and healthcare access, and are disproportionately affected by road traffic injuries due to socioeconomic, geographic, or structural vulnerabilities, limited access to safety infrastructure, protective gear, legal protections, emergency services, Such as informal workers, delivery riders, low income population and rural or peri urban residents, which has been simplified and added to the manuscript for clarity.

Line 77, 78: It is mentioned 28 classes, specify what classes are. ‘Participation was voluntary and gave informed consent ‘check the grammar of this statement

• Line 78 the line has been corrected

• Line 77: the details of the 28 classes had been added:

• The study included a total of 28 classes across four colleges—Kathmandu Medical College and Nepal medical college (Medicine ( 1st, 2nd, 3rd 4th year, Dental (, 2nd, 3rd 4th), BSc Nursing , 2nd, 3rd), College (Developmental Studies, (, 2nd, 3rd 4th) Finance ( , 2nd, 3rd 4th), and St. Xavier’s College (Bachelor in Social Work, , 2nd, 3rd 4th)—with students enrolled from first to fourth year in each stream

Methodology: Add timeline/ duration of the study

• The data collection was done from December 2020 to February 2021, and it has been added in the manuscript

Results: Line 168, Check the grammar. Table 23 appears before table 4. Where are table 2 and 3?

• The errors in the number of the tables have been corrected. Table 23 is actually table 2 and table 4 became table 3.

Discussion:

Line 198 and 199: ‘no statistically significant association

was found between licensing status and RTI occurrence’. This is not reflected in results.

• The aspects of driving behavior and licensing will be addressed in a separate manuscript derived from the same thesis. Therefore, the paragraph discussing license usage will be removed from the current paper to avoid overlapping.

‘The themes and recommendations from the qualitative study’ should be better discussed in this section. The discussion could be strengthened further through more robust comparisons with national and international studies, which would help contextualize the findings, highlight unique challenges in the Nepali context, and support evidence-based recommendations.

• The discussion has been changed, it now includes comparison with national and international studies and reports, and the discussion now holds themes and recommendations from both quantitative and qualitative data.

---

## [Decision Letter · Decision Letter 1]

16 Mar 2026

PONE-D-25-42947R1Motorization and Its Consequences: A Mixed-Methods Study on the Epidemiology, Impact, and Stakeholder Perspectives of Road Traffic Injuries Among Undergraduate Two-Wheeler Riding Students in NepalPLOS One

Dear Dr. Bhattarai,

Thank you for submitting your manuscript to PLOS ONE. After careful consideration, we feel that it has merit but does not fully meet PLOS ONE’s publication criteria as it currently stands. Therefore, we invite you to submit a revised version of the manuscript that addresses the points raised during the review process.

We look forward to receiving your revised manuscript.

Kind regards,

Emma Campbell, PhD

Associate Editor

PLOS One

On behalf of:

Satabdi Mitra, M.D(Community Medicine )

Academic Editor

PLOS One

Journal Requirements:

Additional Editor Comments:

Thank you for making revisions to your manuscript in response to Reviewer 3 comments, unfortunately the wrong decision letter was sent when requesting these changes, as such we are re-issuing the decision to give you the opportunity to formally respond to the concerns and provide the marked up version of the manuscript to show the changes you have made in response to Reviewer 3. We apologise for the inconvenience caused.

Reviewer's Responses to Questions

**Comments to the Author**

1. If the authors have adequately addressed your comments raised in a previous round of review and you feel that this manuscript is now acceptable for publication, you may indicate that here to bypass the “Comments to the Author” section, enter your conflict of interest statement in the “Confidential to Editor” section, and submit your "Accept" recommendation.

Reviewer #1: All comments have been addressed

Reviewer #3: All comments have been addressed

2. Is the manuscript technically sound, and do the data support the conclusions?

Reviewer #1: Yes

Reviewer #3: Partly

3. Has the statistical analysis been performed appropriately and rigorously? 

Reviewer #1: Yes

Reviewer #3: No

4. Have the authors made all data underlying the findings in their manuscript fully available?

Reviewer #1: Yes

Reviewer #3: No

5. Is the manuscript presented in an intelligible fashion and written in standard English?

Reviewer #1: Yes

Reviewer #3: No

6. Review Comments to the Author

Reviewer #1: The presentation has improved a lot at the same time - follow-up publication can not be justification for clarity of this manuscript . further the conclusion and recommendation are general but not restricted to your findings - you can also suggest for better clinical facility to reduce morbidity and disability among affected .

Reviewer #3: Abstract: 5 key recommendations that came forward; change it it 5 key recommendations that emerged

Line 79: Qualitative study was conducted; is it qualitative or quantitative, because rest of the paragraph describes the quantitative data collection process.

Line 82-84: very colloquial

Line 85-86: The data 85 was collected from students enrolled in first to fourth year across 28 academic classes across four colleges, Kathmandu Medical College, Nepal Medical College, College, reword to remove the word appearing across twice.

Line 95-99: It appears that the qualitative and quantitative data were collected from different populations. Additionally, the qualitative component involved only five participants; however, the process by which data saturation was determined is not clearly described. Although the study is presented as a mixed-methods design, the Methods section lacks clarity regarding the specific mixed-methods design employed, the sampling strategies used for each component & in the perspective of mixed methods research, and how the qualitative and quantitative strands were integrated.

Data Analysis:

line 101-107: The data analysis software used both for qualitative and quantitative data analysis to be specified.

Results:

Line 112: was 1878, this number is 1879 in the methods section, (line number 89), be consistent with this data.

Line 114: and 65.6.6 (what is this two decimal number to be clarified)

What does table 1 describe, the table title should be appropriate, if it is of respondents then the n=1217, if it is the one using two wheeler, the findings reveal 578 (line 113), but the table has n=579, to provide titles for all the tables.

Line 154: The expansion of abbreviation NPR is not found in the prior to the usage

Table 3: The themes identified in the qualitative analysis need to be described in greater detail to enhance better understanding. Themes and related verbatim can be jotted together. However, the emerged recommendation require additional description and details with the base for this.

Qualitative data/verbatims clearly lack saturation.

For clarity and uniformity, the Results section should report categorical variables using the n (%) format.

The Results section shows inconsistency in the tense used for reporting findings and requires revision to ensure uniform tense throughout

Discussion:

line 184: This study applied the Safe Systems Approach, and the this approach was not introduced in the background or methodology, and all these need to be consistently aligned throughout.

RTIs/RTI need not be expanded in each section

References: 36, is not found cited

Carefully review the manuscript for typographical errors, punctuation issues (including spacing and use of periods), and inconsistent capitalization between sentences.

7. PLOS authors have the option to publish the peer review history of their article (what does this mean?). If published, this will include your full peer review and any attached files.

Reviewer #1: **Yes:**Dr Shrinivasa B M

Reviewer #3: No

---

## [Author Response · Author response to Decision Letter 2]

17 Apr 2026

Please find all the point by point response to the reviewer’s comments below:

Reviewer #2: REVIEW

• Abstract : Line 29: Methodology: to check this statement ’Methodology: The quantitative section was conducted among undergraduate students in the four colleges in Kathmandu district.’ Quantitative section?

The line has been corrected to qualitative study

• Introduction: Line 52 and 54: ‘Injuries are disproportionately high

among males, disadvantaged groups,’ add why they are disadvantaged groups.

This line is cited from WHO Powered Two- and Three-Wheeler Safety Manual (2022), and it refers to populations who face systemic barriers to safety and healthcare access, and are disproportionately affected by road traffic injuries due to socioeconomic, geographic, or structural vulnerabilities, limited access to safety infrastructure, protective gear, legal protections, emergency services, Such as informal workers, delivery riders, low income population and rural or peri urban residents, which has been simplied and added to the manuscript for clarity with reference.

• Line 77, 78: It is mentioned 28 classes, Specify what are classes. ‘ Participation was voluntary and gave informed consent ‘ check the grammar of this statement

Line 78 the line has been corrected

• Line 77 : the details of the 28 classes had been added :

The study included a total of 28 classes across four colleges—Kathmandu Medical College and Nepal medical college (Medicine ( 1st, 2nd, 3rd 4th year, Dental (, 2nd, 3rd 4th), BSc Nursing , 2nd, 3rd), College (Developmental Studies, (, 2nd, 3rd 4th) Finance ( , 2nd, 3rd 4th), and St. Xavier’s College (Bachelor in Social Work, , 2nd, 3rd 4th)—with students enrolled from first to fourth year in each stream

• Methodology: Add timeline/ duration of the study

The data collection was done from December 2020 to February 2021.

• Results: Line 168, Check the grammar. Table 23 appears before table 4. Where are table 2 and 3?

The errors in the numbering of the tables has been corrected, The table 23 is actually table 2 and table 4 has now been corrected labelled as table 3.

• Discussion:

Line 198 and 199: ‘no statistically significant association

was found between licensing status and RTI occurrence’. This is not reflected in results.

The aspects of driving behavior and licensing are being addressed in a separate manuscript derived from the same thesis. Therefore, the paragraph discussing license usage will be removed from the current paper as it has not been mentioned in the manuscript earlier.

Although licensing was a part of the thesis, it has not been mentioned earlier in the current manuscript. Therefore, to maintain consistency and avoid introducing unsupported findings, the paragraph discussing license usage has been removed.

• ‘The themes and recommendations from the qualitative study’ should be better discussed in this section. The discussion could be strengthened further through more robust comparisons with national and international studies, which would help contextualize the findings, highlight unique challenges in the Nepali context, and support evidence-based recommendations.

The discussion has been fully changed, it now includes comparison with national and international studies and reports and the discussion now holds themes and recommendations from both quantitative and qualitative data.

Please find all the point by point response to the reviewer’s comments below:

Reviewer #3:

• The presentation has improved a lot at the same time - follow-up publication can not be justification for clarity of this manuscript .

-Thank you for your thoughtful feedback. In this study, we primarily focused on the epidemiology and impact of road traffic injuries among undergraduate two wheeler riders. While infrastructure, environmental, and behavioral factors such as road condition, traffic density, time and location of incidents, atmospheric conditions, mechanical aspects of two wheelers, and driving behaviors (speeding, alcohol use, passenger distraction, helmet use, fatigue, extended riding duration, riding while unwell, carrying heavy loads, and traffic law obedience) were explored during the original thesis work, they have not been included in this manuscript due to space constraints. The current paper was already extensive, and we prioritized clarity by limiting the scope to the most relevant findings.

• further the conclusion and recommendation are general but not restricted to your findings - you can also suggest for better clinical facility to reduce morbidity and disability among affected .

-The Conclusion and the recommendations have been changed to go beyond the findings of this paper and incorporated recommendations related to better facilities to reduce disability.

Reviewer #3:

Abstract: 5 key recommendations that came forward; change it it 5 key recommendations that emerged

-This has been addressed in the manuscript

Line 79: Qualitative study was conducted; is it qualitative or quantitative, because rest of the paragraph describes the quantitative data collection process.

-The error has been corrected was describing the quantitative data collection process.

Line 82-84: very colloquial.

-This has been addressed in the manuscript

Line 85-86: The data 85 was collected from students enrolled in first to fourth year across 28 academic classes across four colleges, Kathmandu Medical College, Nepal Medical College, College, reword to remove the word appearing across twice.

-This has been addressed in the manuscript

Line 95-99: It appears that the qualitative and quantitative data were collected from different populations. Additionally, the qualitative component involved only five participants; however, the process by which data saturation was determined is not clearly described. Although the study is presented as a mixed-methods design, the Methods section lacks clarity regarding the specific mixed-methods design employed, the sampling strategies used for each component & in the perspective of mixed methods research, and how the qualitative and quantitative strands were integrated.

Thank you for this observation. The study employed an explanatory sequential mixed methods design, which is a two phase approach. In the first phase, quantitative data were collected from undergraduate students to determine the prevalence of road traffic injuries and associated risk factors. In the second phase, qualitative data were gathered through five key informant interviews with experienced stakeholders (e.g., college coordinator, emergency physician, traffic police officer, former Director General of Transport Management, and judge). These interviews were not intended to achieve thematic saturation in the traditional sense, but rather to capture expert perspectives that could contextualize and extend the quantitative findings.

The sampling strategies therefore differed by strand: the quantitative component used student participants to establish prevalence and risk factors, while the qualitative component purposively selected experts to provide recommendations and system level insights. Integration occurred through the explanatory sequential design, where qualitative findings were explicitly used to interpret and build upon the quantitative results, thereby strengthening the overall interpretation and providing actionable guidance.

Data Analysis:

line 101-107: The data analysis software used both for qualitative and quantitative data analysis to be specified.

-This has been addressed in the manuscript

Results:

Line 112: was 1878, this number is 1879 in the methods section, (line number 89), be consistent with this data.

-This has been addressed in the manuscript, the number is 1879. The consistency has been checked throughout the manuscript.

Line 114: and 65.6.6 (what is this two decimal number to be clarified)

-It has been corrected, it was a typing error

What does table 1 describe, the table title should be appropriate, if it is of respondents then the n=1217, if it is the one using two wheeler, the findings reveal 578 (line 113), but the table has n=579, to provide titles for all the tables.

-This has been addressed in the manuscript, the number is 579. The consistency has been checked throughout the manuscript.

Line 154: The expansion of abbreviation NPR is not found in the prior to the usage

-It was been expanded.

Table 3: The themes identified in the qualitative analysis need to be described in greater detail to enhance better understanding. Themes and related verbatim can be jotted together. However, the emerged recommendation require additional description and details with the base for this.

Qualitative data/verbatims clearly lack saturation.

-I would like to clarify that the qualitative component of this study was explicitly designed to elicit recommendations for reducing road traffic injuries from key stakeholders. The recommendations that emerged were subsequently organized into five thematic areas : education and awareness, reducing dependency on two wheelers, strengthening the role of traffic police, legal reforms, and licensing systems. Within each theme, the recommendations are supported by verbatim quotes from informants to illustrate the rationale and provide context for the suggestions. We acknowledge that the qualitative sample was limited to five informants, and therefore the verbatims may not demonstrate full saturation. However, the intent of this phase was not to achieve thematic saturation in the traditional sense, but rather to capture expert perspectives that complement the quantitative findings. The integration of these two strands follows the explanatory sequential mixed methods design, where the qualitative insights build directly upon the quantitative results to provide depth and actionable guidance.

For clarity and uniformity, the Results section should report categorical variables using the n (%) format.

The Results section shows inconsistency in the tense used for reporting findings and requires revision to ensure uniform tense throughout

-This has been revised thoroughly and has been uniform in reporting categorical variable has been done in n(%) format.

Discussion:

line 184: This study applied the Safe Systems Approach, and the this approach was not introduced in the background or methodology, and all these need to be consistently aligned throughout.

-I agree with you, the safe system approach was not consistently aligned so it has reworked upon.

RTIs/RTI need not be expanded in each section : it has been corrected

References: 36, is not found cited

-It has now been cited

---

## [Editor Report · Decision Letter 2]

24 Apr 2026

Motorization and Its Consequences: A Mixed-Methods Study on the Epidemiology, Impact, and Stakeholder Perspectives of Road Traffic Injuries Among Undergraduate Two-Wheeler Riding Students in Nepal

PONE-D-25-42947R2

Dear Dr.Nikita Bhattarai,

We’re pleased to inform you that your manuscript has been judged scientifically suitable for publication and will be formally accepted for publication once it meets all outstanding technical requirements.

Kind regards,

Satabdi Mitra, M.D(Community Medicine )

Academic Editor

PLOS One
---

## [Editor Report · Acceptance letter]

PONE-D-25-42947R2

PLOS One

Dear Dr. Bhattarai,

I'm pleased to inform you that your manuscript has been deemed suitable for publication in PLOS One. Congratulations! Your manuscript is now being handed over to our production team.

Kind regards,

on behalf of

Dr Satabdi Mitra

Academic Editor

PLOS One